# High-resolution mapping of vehicle emissions of atmospheric pollutants based on large-scale, real-world traffic datasets

Daoyuan Yang [1, #], Shaojun Zhang [1, 2, #], Tianlin Niu [1, 3], Yunjie Wang [1], Honglei Xu [5], K. Max Zhang [2], Ye Wu [1, 4, *]

[1]School of Environment, State Key Joint Laboratory of Environment Simulation and Pollution Control, Tsinghua University, Beijing 100084, P R China.

[2]Sibley School of Mechanical and Aerospace Engineering, Cornell University, Ithaca, NY 14853, U.S.A.

[3]Ricardo Energy & Environment, Beijing 100028, P R China.

[4]State Environmental Protection Key Laboratory of Sources and Control of Air Pollution Complex, Beijing 100084, P R China.

[5]Transport Planning and Research Institute, Ministry of Transport, Beijing 100028, PR China

[#] These authors contributed equally to this paper.

*Correspondence to:* Y. Wu (ywu@tsinghua.edu.cn)

**Abstract**. On-road vehicle emissions are a major contributor to elevated air pollution levels in populous metropolitan areas. We developed a link-level emissions inventory of vehicular pollutants, called EMBEV-Link, based on multiple datasets extracted from the extensive road traffic monitoring network that covers the entire municipality of Beijing, China (16,400 $km^2$). We employed the EMBEV-Link model under various traffic scenarios to capture the significant variability in vehicle emissions, temporally and spatially, due to the real-world traffic dynamics and the traffic restrictions implemented by the local government. The results revealed high carbon monoxide (CO) and total hydrocarbon (THC) emissions in the urban area (i.e., within the Fifth Ring Road) and during rush hours, both associated with the passenger vehicle traffic. By contrast, considerable fractions of nitrogen oxides ($NO_X$), fine particulate matter ($PM_{2.5}$) and black carbon (BC) emissions were present beyond the urban area, as heavy-duty trucks (HDTs) were not allowed to drive through the urban area during daytime. The EMBEV-Link model indicates that non-local HDTs could for 29% and 38% of estimated total on-road emissions of $NO_X$ and $PM_{2.5}$, which were ignored in previous conventional emission inventories. We further combined the EMBEV-Link emission inventory and a computationally efficient dispersion model, RapidAir®, to simulate vehicular $NO_X$ concentrations at fine resolutions (10 m × 10 m in the entire municipality and 1 m × 1 m in the hotspots). The simulated results indicated a close agreement with ground observations and captured sharp concentration gradients from line sources to ambient areas. During the nighttime when the HDT traffic restrictions are lifted, HDTs could be responsible for approximately 10 μg m$^{-3}$ of $NO_X$ in the urban area. The uncertainties of conventional top-down allocation methods, which were widely used to enhance the spatial resolution of vehicle emissions, are also discussed by comparison with the EMBEV-Link emission inventory.

## 1 Introduction

The rapid growth in vehicle use associated with socioeconomic development has triggered serious atmospheric pollution and adverse health impacts (Anenberg et al., 2017; Guo et al., 2014; Huang et al., 2014). Serious air pollution problems, which are seen as high ambient concentration levels of major air pollutants, have raised substantial public attentions in populous metropolitan areas. Beijing implies two profound aspects in one single city: an obvious achievement in city development accompanied by substantial pressure to mitigate air pollution episodes (UNEP, 2016). Many other megacities are facing with similar environmental challenges after decades of rapid economic development. Beijing's annual concentration of fine particulate matter ($PM_{2.5}$) in 2017 was 58 μg m$^{-3}$. Although this value was reduced by 35% as opposed to that in 2013, it still significantly exceeded the limit of China's national ambient air quality standard (35 μg m$^{-3}$) by 66% (Beijing MEEB, 2018a). The recent official source apportionment results indicated that vehicle emissions remained as one of the most important pollution contributors, responsible for an average of 45% of total $PM_{2.5}$ concentrations from local sources (Beijing MEEB, 2018b). The exceedance of ambient nitrogen dioxide ($NO_2$) concentrations represents another air quality problem in Beijing (UNEP, 2016; Beijing MEEB, 2018a), where nitrate aerosols have become one of the most important $PM_{2.5}$ components, with an average mass fraction of up to 40% (Beijing MEEB, 2018b; Li et al., 2018). Therefore, controlling vehicle emissions is one of the prioritized tasks remaining for local environmental protection authorities.

Beijing has been playing a role of pioneer in controlling vehicle emissions within China over the past two decades (Zhang et al., 2014b). So far, emission standards for new vehicles in Beijing have been tightened to the fifth generation (China 5/V standards), and ultra-low sulfur gasoline and diesel fuels have been fully delivered. In addition, after witnessing the effectiveness of driving restrictions (i.e., the "odd-even" policy) to control vehicle emissions during the 2008 Olympic Games, transportation management has been substantially implemented for environmental purposes, notably through license control and driving restriction policies. Currently, traffic measures are increasingly important in the "vehicle-fuel-road" integrated emission mitigation strategies (Wu et al., 2017). For example, the Beijing municipal government has finalized an "Emergency Plan for Extreme Air Pollution in Beijing", which requires issuance of a red alert when a severe pollution episode (e.g., 24 h average concentration of $PM_{2.5}$ above 250 μg m$^{-3}$) lasting over three days is reported. During the red alert periods, private vehicles are prohibited from roads every other day based on the last digit of the license plate, namely, according to the "odd-even" policy.

Although previous testing results could convincingly support the decreasing trend in fleet-average emission factors for local vehicles (Zhang et al., 2014b; Wu et al., 2012), some major limitations have not yet been adequately addressed. First, a major aspect of previous assessment tools, known as emission inventories (et al., Lang et al., 2012; Zhang et al., 2014b), was

developed based on vehicle registration data lacking temporal and spatial associations with real-world traffic patterns. Only a few studies have attempted to establish cell-gridded or link-based emission inventories that limited their study domains to the urban area (e.g., within the Fifth or Sixth Ring Roads) and/or limited vehicle categories (e.g., light-duty passenger vehicles) (Huo et al., 2009; Wang et al., 2009; Jing et al., 2016). Nevertheless, the total municipal area of Beijing is approximately 16,400 km$^2$, and vehicular emissions in the outskirts should be evaluated. As a regional transportation hub, it is known that a considerable of freight trucks registered in other regions are operated within the city boundary of Beijing. All previous studies have not quantified on-road emissions from non-local trucks. On-road measurement studies using a plume chasing method indicated that non-local trucks were highly likely to be gross emitters of primary PM$_{2.5}$ and black carbon (BC) (Wang et al., 2012), since their original registration regions usually were less strict with respect to environmental oversights (e.g., type-approval conformity check, in-use compliance inspection) than Beijing (Zheng et al., 2015).

Driven by the rapid development of intelligent transportation systems (ITS) in many cities during recent decades, we are able to collect real-world traffic data by multiple ITS approaches (Barth, 2003; Gately et al., 2017; Zhang et al., 2018). These ITS informed datasets are capable of capturing the dynamic traffic conditions in congested urban areas as well as actual driving patterns of diesel trucks, which could contribute large fractions of NO$_X$ and PM$_{2.5}$ despite small vehicle numbers (Dallmann et al., 2013; Gately et al., 2017). In this study, we established a high-resolution emission inventory of on-road vehicles (EMBEV-Link) based on large-scale, real-world traffic datasets (e.g., traffic count, hourly speed, fleet configuration), which covered the entire road network of the municipality of Beijing. This tool enabled us to elucidate the temporal and spatial emission patterns and to detail the emission burden from non-local trucks. This paper presents an example to conduct fine-grained emission modeling at the megacity scale and can directly support local emission mitigation strategies.

## 2 Methodology and Data

### 2.1 Research domain and emission calculation

The entire municipality of Beijing, with a total area of 16,400 km$^2$, comprises sixteen urban, suburban, and rural districts. The present city progressively spreads outwards in concentric ring expressways (i.e., Second to Sixth Ring Roads). The urban area is typically referred to as the region within the Fifth Ring Road, wherein the municipal government has intensively implemented driving restrictions since 2008. Emissions of primary vehicular pollutants (carbon monoxide, CO; total hydrocarbon, THC; nitrogen oxide, NO$_X$, PM$_{2.5}$, and black carbon, BC) were calculated with a high-resolution method in a temporal and spatial framework, namely, the Link-level Emission Model for BEijing Vehicle Fleet (EMBEV-Link). Fig. 1 is a flow diagram to illustrate the overall modeling methodology for the EMBEV-Link system. The traffic data acquisition and

further modeling to the entire road network will be introduced detailed in Section 2.2. For each road link, hourly emissions are the product of traffic volume, link length and speed-dependent emission factors (see Eq. 1) (Zhang et al., 2016).

$$E_{h,j,l} = \sum_t EF_{c,j}(v) \times TV_{c,h,l} \times L_l \tag{1}$$

where $E_{h,j,l}$ is the total emission of pollutant $j$ on road link $l$ at hour $h$, units in g h$^{-1}$; $EF_{c,j}(v)$ is the average emission factor of pollutant $j$ for vehicle category $c$ at speed $v$, units in g km$^{-1}$; $TV_{c,h,j}$ is the traffic volume of vehicle category $c$ on road link $l$ at hour $h$, units in veh h$^{-1}$; $L_l$ is the length of road link $l$, units in km. Eight vehicle categories were defined, namely, light-duty passenger vehicle (LDPV), medium-duty passenger vehicle (MDPV), heavy-duty passenger vehicle (HDPV), light-duty truck (LDT), heavy-duty truck (HDT), public bus, taxi and motorcycles (MC) (See Table S1). For HDTs, we further classified into local HDTs and non-local HDTs according to the registration location. Significantly higher BC emission factors were identified from non-local HDTs than from local HDTs because Beijing has more stringent conformity enforcement requirements (Wang et al., 2012).

The speed-dependent emission factors for each vehicle category were developed based on the official Emission Factor Model for the Beijing Vehicle Fleet Version (EMBEV 2.0). The EMBEV model was developed based on thousands of in-lab dynamometer tests and hundreds of on-road tests (Zhang et al., 2014b). Now, the EMBEV methodology and key parameters have been essentially referred to by China's National Emission Inventory Guidebook (Wu et al., 2016; Wu et al., 2017). Fig S1 presents speed-dependent emission factors of CO, NO$_X$ and BC for LDPV and HDT categories representing average environmental conditions, fleet configurations (e.g., fuel type, emission standard and vehicle size) and fuel quality (e.g., sulfur content). To match the traffic data, we utilized 2013-2014 as the calendar year to estimate emission factors. The original EMBEV model included evaporative THC emissions for gasoline vehicles (Zhang et al., 2014b). Later on, we revised the diurnal and hot soak emission rates based on local SHED tests (Liu et al., 2015) and estimated that the evaporative THC emissions could be responsible for approximately 30% of total THC emissions in Beijing. In the current EMBEV-Link work, evaporative THC emissions were not included because we are limited to spatially specifying the evaporative off-network emissions. Furthermore, air quality simulations often require non-methane volatile organic compounds (NMVOC) as emission input to simulate secondary pollutant formation (e.g., ozone and secondary organic aerosol). Based on local dynamometer measurements, NMVOC could approximately account for 90% of tailpipe THC emissions. The detailed species-resolved measurement profiles are more sensitive to vehicle technologies, fuel properties and environmental conditions, which should be developed based on advanced measurements.

## 2.2 Generating dynamic traffic profiles based on real-time congestion information

High-resolution congestion mapping was developed based on densely distributed taxis (more than 60,000 vehicles) in Beijing, known as floating cars (Cai and Xu, 2013). The municipal traffic commission used numerous trajectory data of GPS-instrumented taxis to estimate color-informed congestion levels (red: serious congestion; orange: moderate congestion; grey: not congested; see Fig S2 for example). The congestion level was defined by real-time speed and was updated every 5 min. Nevertheless, although dynamic traffic conditions were visualized by congestion maps, most required data such as link-level

speeds were not available. From the official website (http://www.bjtrc.org.cn/), the only open-accessed data in addition to real-time congestion maps were hourly speeds and congestion indexes for ring expressway-defined traffic regions. To improve the spatial resolution, we developed an image recognition program to parameterize the congestion level based on available congestion maps (141 available days annually in this study). Furthermore, link-level hourly speeds were calculated based on the relationship between congestion index and average speed. The calculation method is documented in the Supplement, Part

II. On the aggregate level, the biases of rush hour speeds between estimated results and reported data were within ±5% for all districts. It is noted that link-level speeds for public buses were corrected due to their frequent stops for discharging and picking up passengers (Zhang et al., 2014a).

We further used congestion map-informed road speeds to improve the temporal resolution of traffic volumes, which were originally investigated on an annual basis (BJTU and Beijing EPB, 2014). Traffic density modeling was used to express the

relationship between total volume and speed in this study. The Underwood-style traffic density models (see Eq. 2) were used for expressways and arterial roads, respectively, which better fit the local traffic profiles than the Greenshields model (Hooper et al., 2013; Wang et al., 2013).

$$q = k_m u \ln \frac{u_f}{u} \qquad (2)$$

where $q$ is the lane-specific traffic volume at speed $u$, veh h$^{-1}$; $u$ is the hourly average traffic speed, units in km h$^{-1}$; and $k_m$ is

the best fitting traffic density, veh km$^{-1}$. The model coefficients, $k_m$ and $u_f$, are determined through linear least squares fitting based on annual-average hourly volume and speed profiles for urban major roads (see the Supplement, Part II). We applied the Underwood model to estimate the relative change of hourly traffic volume in response to the speed variation.

Traffic video records were collected at more than 30 major urban roads to develop traffic mixes by hour, road type and district (see Fig. S3). In particular, we manually counted the amount of non-local HDTs at the representative road sites and

distinguished volume allocations for local and non-local traffic during different hours during the night (Zhang et al., 2017). The suburban and rural areas outside of the Fifth Ring Roads were scarcely covered by municipal floating cars and traffic investigations. The Ministry of Transport has established a nationwide networking to monitor intercity traffic conditions

(Zhang et al., 2018). Twenty-four-hour diurnal traffic profiles including volume, speed and fleet mix were obtained from 70 highway sites in Beijing, leading to an improved understanding of traffic patterns in the outlying districts beyond the Fifth Ring Road. Taking G-101 as an in-depth example (See Fig. S4), apparent morning and evening peaks were observed at one site (Site A) close to the North Sixth Ring Road, representing urban passenger travel demand. By contrast, HDTs were responsible for nearly half of the total volume at one remote site approaching the municipal border (Site B) and peaked around noon. Supplement Part II also includes the technical details regarding estimating traffic profiles for nonmonitored roads (e.g., residential roads).

**2.3 Traffic scenarios under various transportation management schemes**

In this study, four scenarios were generated as inputs for the EMBEV-Link to observe the impacts from major transportation management schemes. Table S2 details the traffic management schemes enforced for major vehicle categories during various traffic scenarios. Scenario Weekday (*S1*) estimated annual-average traffic patterns during weekdays (Monday to Friday) with regular driving restriction rules on personal car use. Scenario Weekend (*S2*) estimated average traffic patterns during weekends (Saturday and Sunday) without regular driving restrictions, when urban residents tend to reduce commutes but increase casual trips. Scenario Congestion (*S3*) reflected the most congested conditions that occasionally existed during the weekends prior to some statutory holidays (e.g., Workers' Day on May $1^{st}$ and National Day on Oct $1^{st}$). During the special weekends, the scheduling program was adjusted according to normal weekdays, but the driving restrictions were not implemented. Scenario APEC (*S4*) estimated the traffic patterns during the Asia-Pacific Economic Cooperation Summit with much stricter traffic limitations than normal situations. Half of all personal vehicles were restricted from roads by the odd-even policy, and non-local trucks were also strictly prohibited from journeying into the city.

**2.4 Dispersion mapping for vehicular pollutants**

The RapidAir® model developed by Ricardo Energy & Environment was combined with EMBEV-Link to simulate vehicular concentrations of $NO_X$ for the entire domain and typical hotspot areas. RapidAir® combines the EPA's Gaussian plume dispersion model (AERMOD) and open-source computing algorithms by using a kernel convolution that creates millions of overlapping plumes from emission sources and sums distance-weighted concentrations at each receptor cell (Masey et al., 2018). Using unified emissions and meteorological inputs, RapidAir® can produce concentration results in strong agreement with other Gaussian dispersion models (e.g., AERMOD, ADMS) while greatly improving computational efficiency (e.g., 5 mins for each hotspot). This study selected $NO_X$ as the simulated pollutant category due to the high contribution from traffic emissions. For the entire municipality, hourly $NO_X$ concentrations contributed by vehicle emissions were simulated at a spatial

resolution of 10 m × 10 m, which used the annual-average hourly meteorological data (e.g., temperature, wind speed, wind direction) as modeling inputs. Two typical hotspots at the Central Business District (Guomao) and along a major suburban freeway (Xisanqi) were selected for more fine-grained simulations. The receptor cells in the hotspot areas were meshed into 1 m × 1 m in order to visualize the $NO_X$ concentration gradients from road, to curbside and thus throughout the ambient urban zone. Detailed data sources and key parameters of meteorological and terrain input profiles are described in the Supplement, Part III.

## 3 Results and Discussion

### 3.1 Traffic and emission patterns under various scenarios

The daily traffic activities during weekdays (*S1*, 258 million veh·km) and weekends (*S2*, 259 million veh·km) are estimated to be close to each other, representing comparable effects from the increased commute travel demand during weekdays and the absence of regular driving restrictions during weekends (See Fig. S5). However, the diurnal fluctuations of average speeds depict different travel characteristics between weekdays and weekends. The two most congested periods with lowest traffic speeds (below 23 km h$^{-1}$) clearly occurred during the mornings (8:00 and 9:00 GMT+8; note: 8:00 hereafter represents the entire hour from 8:00 to 8:59 GMT+8) and evenings (18:00 and 19:00 GMT+8) of weekdays. By contrast, we could not observe similar morning traffic peaks during weekends, but traffic conditions deteriorated throughout the afternoon (15:00 to 18:00 GMT+8), reflecting frequent casual travels. Combined with the daily traffic activity of *S1* and *S2*, we could calculate the annual vehicle kilometer travelled (VKT) in Beijing. For all vehicle categories except HDPVs, the EMBEV-Link indicated that VKT data showed good agreement (i.e., relative bias within ±6%) with the results derived from the official vehicle inspection database (See Fig. S6). The remaining excess of estimated annual VKT of the HDPVs is probably contributed by non-local HDVPs, whose emissions are not estimated in a separate vehicle category.

Two special scenarios (*S3* and *S4*) indicate the substantial impacts from municipal transportation management on traffic activities in Beijing. Without strict driving restrictions, the 24 h average speed within the Fifth Ring Road decreased to merely 23 km h$^{-1}$ under S3 (See Fig. S7), indicating that the daily level of congestion was almost comparable to the rush hours of normal weekdays. The daily traffic activity was then increased by 8% versus that of normal weekdays. By contrast, the odd-even policy was implemented during the APEC Summit week and played an effective role in reducing traffic demand and alleviating road congestion. The daily traffic activity under *S4* was lowered by 12%, while the average speed rose to 35 km h$^{-1}$. It is noted that additional control actions were simultaneously enforced upon heavy-duty trucks during the APEC period, which did not significantly change overall traffic patterns compared with the strictly controlled LDPV fleet but greatly

contributed to emission reductions (see next section).

Total daily emissions estimated by the EMBEV-Link model are 823 tons for CO, 84.4 tons for THC, 326 tons for $NO_X$, 10.6 tons for $PM_{2.5}$ and 5.5 tons for BC, respectively, during weekdays (*S1*, See Fig. 2). During weekends (*S2*), total vehicle emissions decreased by small percentages (e.g., 3% for CO and THC, 5% to 7% for $NO_X$, $PM_{2.5}$ and BC). Greater traffic demand and more serious congestion under *S3* combined to trigger increased vehicle emissions, e.g., 12% for CO and THC and 6% for $NO_X$, $PM_{2.5}$ and BC in the entire municipality. The CO and THC emission enhancements were more significant in the urban areas, increased by 17% compared with *S1*, representing the effect from the increasing amount of on-road LDPVs during the special period. The recent traffic monitoring data indicate the overall congestion in the urban area has not changed significantly, which is owing to the stringent restrictions on the registration of new vehicles in Beijing (BTI, 2018). On the other hand, average emission factors have decrease significantly due to the implementation of newer emission standards and the subsidized scrappage of older vehicles. As a result, we estimated that the total daily emissions would be 523 tons for CO, 62.5 tons for THC, 256 tons for $NO_X$, 8.33 tons for $PM_{2.5}$ and 4.18 tons for BC, respectively, in 2017. The significant reductions are primarily attributed to the improvements in average vehicle emission factors.

Comprehensive traffic controls are estimated to greatly reduce total vehicle emissions by 43% for CO, 44% for THC, 28% for $NO_X$, 37% for $PM_{2.5}$ and 35% for BC under *S4* relative to *S1*. The greater reductions in CO and THC resulted from the greatly increased average speeds of urban LDPVs and taxis, resulting in lower emission factors. However, diesel freight trucks were responsible for a major part of $NO_X$, $PM_{2.5}$ and BC emissions. More than 80% of total traffic activities for HDTs were distributed beyond the Fifth Ring Road, where traffic congestion was less serious and emission factors were less sensitive. However, the Beijing municipal government dispatched more public buses for transportation services during the APEC period (Beijing Municipal Government, 2014), which would increase $NO_X$ emissions as opposed to the normal bus fleet. Overall, the average concentration of $NO_2$ during the APEC period was 46 $\mu g\ m^{-3}$, representing a reduction of 31% compared with the same period of the prior year (Beijing EPB, 2014). The air quality benefit with respect to ambient $NO_2$ concentrations was in line with the comparative results between *S1* and *S4*.

**3.2 Temporal and spatial characteristics of vehicle emissions**

The major temporal difference in emission patterns between *S1* and *S2* is higher emissions during weekday rush hours. We thus refer to the weekday scenario (*S1*) to elucidate temporal and spatial emission patterns (see Figs. 3 and 4). For CO and THC, their emission peaks during morning (7:00 to 9:00 GMT+8) and evening (17:00 to 19:00 GMT+8) rush hours are apparently associated with diurnal fluctuations in passenger travel demand. For example, the highest hourly emissions of CO

and THC were estimated during the morning rush hour (7:00 GMT+8), higher than the 24 h averages by approximately 90%. As Figs. 4a and 4b visualize, CO emission intensity in the urban area is significantly higher than that in the outlying area during both peak and nighttime periods. Table 1 summarizes the emissions allocation by vehicle categories and regions according to EMBEV-Link. The emission allocations show high resemblance between CO and THC: 55-60% of city-total emissions are estimated to exist within the urban area, where LDPVs and taxis dominate the contributions. CO and THC emissions also exhibit heterogeneously diurnal fluctuations in various traffic regions (see Fig. 3) because they are both primarily contributed by LDPVs which comply with typical two-peak patterns on the whole.

Diesel fleets (e.g., HDTs, HDPVs, Bus) are responsible for much greater shares of the vehicle emissions of $NO_X$, $PM_{2.5}$ and BC compared with their contributions to CO and THC. Consequently, distinctive traffic behaviors of these diesel fleets would result in disparate temporal and spatial emission patterns than those for CO and THC, which are more significantly influenced by gasoline fleets. First, we could additionally observe elevated total emissions of $NO_X$, $PM_{2.5}$ and BC after 11 pm and during the nighttime period (2:00 to 4:00 GMT+8, Fig. 3), which are not discerned from CO and THC emission patterns. These elevated emissions are caused by the local traffic restrictions for HDTs during the daytime, which would activate the HDT traffic during nighttime hours (after 23:00 GMT+8). Second, nearly 70% of $NO_X$, $PM_{2.5}$ and BC emissions occur outside the urban area (see Fig. 2), and the emission contributions of local HDTs and non-local HDTs account for the largest proportion (approximately 70% to 80%, see Table 1). By contrast, the public buses contribute 16% of the total $NO_X$ emissions and 7% of the total $PM_{2.5}$ emissions in the entire city; in the urban area, buses contribute 30% of $NO_X$ emissions (see Table 1). The EMBEV-Link emission maps indicate that many HDTs would likely flood into the urban area during the midnight period, leading to higher emissions on major urban roads (e.g., urban Ring expressways) (Fig. 4c); however, these HDTs would travel between the Fifth and Sixth Ring Roads or on other outlying expressways during the daytime period (Fig. 4d). In Section 3.5, we further explore the environmental impacts contributed by these diesel trucks.

Intra-day variability of traffic conditions during the same hour has been observed based on available traffic monitoring data, which can further impact traffic emissions. Using two urban roads as examples (West Third Ring Rd. and Zizhuqiao Rd.), we developed the distributions of inter-day hourly speeds during a typical morning rush duration (8:00 GMT+8) (see Fig. S8). Although the speed distributions for the expressway (West Third Ring Rd.) and sub-arterial (Zizhuqiao Rd.) representatives show various patterns, the input data applied in *S1* are close to the mean values of speed distributions. Furthermore, despite of inter-day variabilities (within ±15% for 95% variation intervals), the estimated emission factors and emission intensities in *S1* also approximate to the mean values of the results during various workdays (bias less than 4%).

### 3.3 High-resolution simulation of vehicular $NO_X$ concentrations

Fig. 5a illustrates the spatial distribution of annual-average $NO_X$ concentrations for each cell (meshed into 10 m × 10 m) contributed by vehicle emissions. Clearly, the spatial variations in the simulated concentrations highly resemble the emission patterns. The cell-average $NO_X$ concentrations within the Sixth Ring Road are simulated as 46.1 μg m$^{-3}$, significantly higher than the level of outlying areas. Beyond the Sixth Ring Road, moderate impacts could also be observed in proximity to several intercity expressways with considerable traffic fractions of HDTs. Two hotspots in close proximity to busy roads, Guomao (Fig 5b) and Xisanqi (Fig. 5c), each have average $NO_X$ concentrations above 100 μg m$^{-3}$. The RapidAir model is capable of visualizing the $NO_X$ decline gradients from road to near-road ambient zone at the two hotspots, as well as the areas surrounded by densely packed buildings influenced by street canyon effects. Extremely high $NO_X$ concentrations are observed in the road environments, which would substantially influence up to 50 m across the expressways (over 200 μg m$^{-3}$), and up to 20 m for the arterial roads (over 100 μg m$^{-3}$) (see the Supplement, Part III).

In China, the environmental protection authorities only report the $NO_2$ concentrations measured at the official air quality monitoring sites, which do not include NO concentrations. We referred to the approximate photostationary state (i.e., chemical equilibrium between the $NO_2$ photolysis and the $O_3$ depletion) to estimate total NO concentrations for the official sites (Yang et al., 2018). In this study, we only used the tropospheric $NO_X$ chemistry to estimate NO concentrations during the daytime (approximately 6:00 to 18:00 GMT+8 as the annual average) and derived the total $NO_X$ as the sum of observed $NO_2$ and estimated NO (see the Supplement, Part III). In Fig. S9, we compared the simulated $NO_X$ concentrations contributed only by vehicle emissions and the total $NO_X$ concentrations for 17 official air quality sites (12 urban sites and 5 traffic sites) (see the Supplement, Part III). First, the significantly strong correlation ($R^2$=0.89) between vehicular $NO_X$ and total $NO_X$ indicates that the EMBEV-Link inventory has reasonably captured the spatial distribution of vehicular $NO_X$ emissions. Furthermore, the average ratios of vehicular $NO_X$ within total $NO_X$ suggest substantial contributions from on-road vehicles: 76% for urban sites and 87% for traffic sites (i.e., daytime annual-average). The remaining portion of $NO_X$ concentrations could be attributed to regional background and other local sources, which account for a minor part compared with traffic emissions. We acknowledge that the daytime concentration of other reactive oxides of nitrogen (i.e., $NO_Z$, including $HNO_3$ and HONO) could be approximately 10% of concurrent $NO_X$ concentrations by analyzing the air quality simulation outputs of Zheng et al. (2019). Further studies would be needed to couple dispersion and advanced atmospheric chemistry to better resolve urban pollution.

In this study, we estimated the $NO_X$ emissions and concentrations both based on annual-average environmental conditions. In addition to seasonal changes of dispersion conditions (e.g., wind speed, wind direction), $NO_X$ emissions could be probably affected by ambient temperature conditions. Of note, by analyzing more than hundreds of thousand European vehicles by using

remote sensing measurements, strong temperature dependence for $NO_X$ emissions of diesel cars has been identified (Grange et al., 2019; Borken-Kleefled and Dallmann, 2018). $NO_X$ emission factors from diesel cars significantly increase during wintertime, which have not been sophisticatedly characterized by many emission models. $NO_X$ emissions from heavy-duty diesel vehicles in Beijing during wintertime could be probably elevated because of similar temperature impacts, which should be carefully characterized by analyzing local measurement data in the future.

## 3.4 The environmental impacts from heavy-duty trucks (HDTs)

Conventional emission inventories were developed based on the registered vehicle population to support on-road emission management in Beijing (Zhang et al., 2014b). However, the significant non-local truck traffic was not reflected by the static registration data. The EMBEV-Link shows that non-local HDTs emitted 2.46 tons of $NO_X$, 1.07 tons of $PM_{2.5}$ and 0.68 tons of BC annually, respectively, which were responsible for 29%, 38% and 47% of estimated total emissions in 2013. The greatest discrepancy of BC further represents higher BC emission factors of non-local HDTs than those for local HDTs. In other words, the previous conventional emission inventory (Zhang et al., 2014b) underestimated the emissions of $NO_X$ and $PM_{2.5}$ by 45.2% and 45.1%, respectively, which was primarily due to the missing contributions from non-local HDTs.

Stringent transportation management for HDTs in Beijing caused their travel behaviors and air pollutant emissions to sharply vary from other vehicle categories, both temporally and spatially. During the daytime with urban restrictions (before 23:00 GMT+8), we could scarcely observe on-road HDTs other than special municipal vehicles within the Fifth Ring Road. Consequently, the total HDT emissions (local and non-local combined) predominantly appeared beyond the Fifth Ring Road (68% of $NO_X$, 70% of $PM_{2.5}$, and 76% of BC), including a considerable fraction in the area between the Fifth Ring Road and Sixth Ring Road (40% of $NO_X$, 39% of $PM_{2.5}$, and 42% of BC). By contrast, many HDTs drove into the urban area during the nighttime without such restrictions. Therefore, we could clearly observe a significant elevation of HDT emissions within the Fifth Ring Road beginning at precisely 23:00 (GMT+8). The RapidAir® model is applied to visualize $NO_X$ concentrations contributed by HDTs exclusively. During the daytime (6:00 to 22:00 GMT+8), HDTs primarily contributed to high concentration spots scattered near major expressways between the Fifth Ring Road and Sixth Ring Road (See Fig. S10). Nevertheless, the nighttime impact (23:00 to 5:00 GMT+8) was more significant due to the concentrated urban truck activity and more unfavorable dispersion conditions (e.g., lower stable boundary layer). Total HDTs could contribute $9.8\pm1.6$ μg m$^{-3}$ of $NO_X$ during the night period, including $6.3\pm1.0$ μg m$^{-3}$ from non-local HDTs. This study has quantified results regarding the air quality impacts from non-local trucks, which is an important issue of air quality management which has been neglected in previous studies (Li et al., 2015). Future studies utilizing this improved emission inventory could include the characterization

of secondary air pollutants contributed by non-local traffic. Managing road freight transportation in Beijing is a regional task, which is controlled even beyond the Jing-Jin-Ji region and is highly relevant to other coal-rich provincial areas (e.g., Shanxi, Inner Mongolia). We suggest that the research domain be enlarged to include these surrounding provincial areas as more traffic data become available in the future.

**3.5 A comparative discussion on various methods to construct link-based emission inventories**

Traffic data availability is a significant challenge in characterizing real-world spatial and temporal distributions of on-road vehicle emissions. As high-resolution emissions are essentially required by air quality simulations, other accessible spatial surrogates are used to artificially allocate total vehicle emissions into fine spatial cells. Population density and/or road length density are two typical varieties of spatial indicators to allocate vehicle emissions by assuming linear relationships between vehicle emissions and spatial surrogates (Zheng et al., 2009; Zheng et al., 2014). However, such top-down allocations are often questioned with respect to the accurate representation of real-world traffic activity. We compare three methods of developing emission inventories with spatial resolutions into 1 km × 1 km gridded cells. M1 denotes this study (EMBEV-Link) using link-level traffic data and reflects real-world emission patterns. M2 and M3 denote two top-down allocation methods based on population density and road length density, respectively (see the Supplement, Part IV). To observe only the effect from using spatial surrogates, estimated total emissions of M1 are also used by M2 and M3 allocations. For M2 allocation, the GIS-based population density is obtained from the LandScan 2012 population database (ORNL, 2012). Standard road length (Zheng et al., 2009), one proxy parameter to further take account of traffic flow distinctions between urban and rural areas, is applied in M3 allocation instead of actual road length. CO and BC are discussed, as they represent gasoline and diesel featured emissions, respectively.

As Fig. 6 illustrates, M2 generates many scattered emission hotspots in accordance with highly populous communities in both urban and suburb/rural areas, but weakly represents the topology of road networks. Compared with M1, M2 tends to underestimate CO emissions in the urban area but overestimates for the outlying areas, because the static population distribution quite differs from actual travel activities. Many people reside outside the Fifth Ring Road, where housing costs are relatively lower, but must travel into the urban area for employment or casual purposes. However, M2 artificially estimates a number of urban hotspots regarding BC emissions (72% of the cells within the Fifth Ring Road are overestimated) while substantially underestimating the emission density between the Fifth and Sixth Ring Roads (68% of the cells in that region are underestimated). Such distortion is caused by the simple assumption of a proportional relationship between BC emissions and population density, as well as the absent accounting of HDT driving restrictions within M2.

M3 reflects the topology of traffic emission as line sources to some extent but underestimates CO emissions within the Fifth Ring Road compared with M1 by 28%. This is because although M3 considers the traffic volume characteristics according to region and road type, increased emission factors due to traffic congestion are not accounted. CO emission density between the Fifth and Sixth Ring Roads is overestimated by M3 because the same coefficients as for the urban area are applied to calculate standard road lengths. In contrast, M3 overestimates BC emissions in the urban area, but underestimates emissions between

the Fifth and Sixth Ring Roads, due to the absent consideration of local traffic restrictions on HDTs. In addition, we also observe that M3 tends to overestimate both CO and BC emissions in the northern areas with intercity expressways; however, underestimations in M3 are identified in the southern areas. This is because M3 considers unified traffic volume weights for all of the intercity highways outside the Sixth Ring Road. In reality, the traffic monitoring data reveal that outlying expressways in the southern areas have greater traffic volumes than the northern expressways, which connect to hilly and less populous

regions.

Currently, secondary aerosols (e.g., nitrate and secondary organic aerosols) are the leading chemical components of $PM_{2.5}$ concentrations in Beijing. Air quality management in this megacity requires fine-grained air quality simulations by improving emission patterns and including chemical transport simulations. As discussed above, the EMBEV-link model enables us to characterize the spatial heterogeneity in real-world traffic emissions to support small-scale simulations (e.g., down to 1 km-

scale), which are typically completed at approximately 3 to 4-km scales in current studies (e.g., Zheng et al., 2019). To better fulfill this function, the seasonal variability and species-resolved NMVOC profiles are also required to be improved in the EMBEV-Link model.

**4 Conclusions**

This study presents the development of high-resolution emission inventory of vehicle emissions in Beijing (EMBEV-Link) by

using multiple large-scale traffic monitoring datasets. Real-time traffic congestion index maps, intercity highway monitoring and manual traffic investigations were applied to estimate link-level and hourly profiles for traffic volume, fleet composition and road speed. We applied the EMBEV-Link model to four typical traffic scenarios in order to elucidate spatial and temporal patterns of vehicle emissions in association with different transportation management schemes in Beijing. The vehicular $NO_X$ concentrations were simulated by using the RapidAir® model at high spatial resolutions, meshed into 10 m × 10 m cells in the

entire municipality and further 1 m × 1 m cells in the hotspots.

The EMBEV-Link results indicate significant impacts on temporal and spatial patterns of vehicle emissions caused by the traffic restrictions in Beijing. Total vehicle emissions were estimated as 823 tons for CO, 84.4 tons for THC, 326 tons for $NO_X$,

10.6 tons for $PM_{2.5}$ and 5.5 tons for BC, respectively, during an average weekday (*S1*) of 2013. CO and THC emissions are featured as pollutants contributed by gasoline vehicles, whose peaks were identified in the urban area and during traffic rush hours. By contrast, $NO_X$, $PM_{2.5}$ and BC were considerably contributed by diesel fleets, whose emissions peaked between Fifth and Sixth Ring Roads during the daytime and then flooded within the Fifth Ring Road when truck restrictions were not implemented. The overall emissions during weekends (*S2*) were close to the weekday levels because the urban traffic restrictions on LDPVs were not enforced during weekends. The absence of regular restrictions on LPDVs would trigger serious congestion and lead to 12% increases of CO and THC emissions in the entire municipality (*S3*), in comparison with the normal weekday levels. On the other hand, the stringent traffic controls implemented during the APEC Summit period (*S4*) could reduce vehicle emissions by approximately 30% to 40%, varying by pollutant category.

We further demonstrated a few major improvements by EMBEV-Link compared with previous emission inventory methods. First, the EMBEV estimated that non-local HDTs contributed 2.46 tons of $NO_X$, 1.07 tons of $PM_{2.5}$ and 0.68 tons of BC annually, respectively, which were responsible for 29%, 38% and 47% of estimated total emissions in 2013. Nevertheless, these emissions from non-local HDTs were missing from the registration-based emission estimates. A considerable fraction of truck traffic would flood into the urban area after 23:00 GMT+8, resulting in approximately 10 μg m$^{-3}$ of nighttime $NO_X$ concentrations there. Second, combined with the RapidAir® model, the link-level emissions could represent a valuable asset to map high-resolution concentrations of vehicular pollutants over a large geographical area. The case study of $NO_X$ dispersion indicated a large contribution from traffic emissions, with strong agreement with observation data in the urban area, and sharp elevation gradients from ambient areas to roads in the hotspots. Finally, we also revealed that the conventional top-down allocation methods according to population or road density could cause significant uncertainties in the spatial distributions of vehicle emissions, because these allocation methods were limited to consider both the real traffic patterns and the effects of local traffic restrictions.

**Author Contribution**

D. Y. and S. Z. contributed equally to this research. S. Z. and Y. Wu conceived the research idea; D. Y., S. Z. and H. X. prepared the traffic dataset; D. Y. contributed to the new emission inventory model; T. N. conducted the dispersion simulation; D. Y., S. Z., Y. Wang, K.M. Z., and Y. Wu analyzed the data; D. Y. and T. N. drew the figures; S. Z., D. Y., and Y. Wu wrote the paper with contributions from all the authors.

**Acknowledgments**

This study was supported by the National Natural Science Foundation of China (91544222) and the National Key Research and Development Program of China (2017YFC0212100). S. Z. is supported by Cornell University's David R Atkinson Center for a Sustainable Future (ACSF Postdoctoral Fellowship). K.M. Z. acknowledges support from the National Science Foundation (NSF) through grant No.1605407. The contents of this paper are solely the responsibility of the authors and do not necessarily represent official views of the sponsors or companies.

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

**Tables and Figures**

**Table 1 Daily emission allocation by vehicle category and region under the weekday traffic scenario (*S1*)**

| Air pollutants | Region | Daily emissions(t/d) | Emission allocation by vehicle category group | | | | |
|---|---|---|---|---|---|---|---|
| | | | LDPV & Taxi | MHDPV & Bus | Local Trucks | Non-local Trucks | Others |
| CO emissions | Within the Fifth Ring Road [a] | 458 | 77.7% | 11.9% | 7.6% | 1.2% | 1.7% |
| | Between the Fifth and Sixth Ring Roads [b] | 233 | 36.1% | 28.1% | 22.6% | 9.7% | 3.6% |
| | Outside the Sixth Ring Roads | 142 | 26.5% | 29.0% | 24.6% | 12.5% | 7.4% |
| THC emissions | Within the Fifth Ring Road | 49.0 | 78.8% | 10.5% | 6.7% | 1.5% | 0.5% |
| | Between the Fifth and Sixth Ring Roads | 21.2 | 37.6% | 24.1% | 19.0% | 14.2% | 5.1% |
| | Outside the Sixth Ring Roads | 14.2 | 27.9% | 24.3% | 19.6% | 17.3% | 10.9% |
| $NO_X$ emissions | Within the Fifth Ring Road | 104.1 | 22.5% | 38.4% | 29.0% | 10.0% | 0.1% |
| | Between the Fifth and Sixth Ring Roads | 130.8 | 5.3% | 24.3% | 35.2% | 35.2% | 0.1% |
| | Outside the Sixth Ring Roads | 91.0 | 3.5% | 17.0% | 37.8% | 41.5% | 0.2% |
| $PM_{2.5}$ emissions | Within the Fifth Ring Road | 3.19 | 25.1% | 31.1% | 28.5% | 15.0% | 0.4% |
| | Between the Fifth and Sixth Ring Roads | 4.17 | 5.7% | 15.6% | 31.1% | 47.4% | 0.3% |
| | Outside the Sixth Ring Roads | 3.30 | 3.7% | 16.4% | 29.6% | 49.8% | 0.5% |
| BC emissions | Within the Fifth Ring Road | 1.34 | 10.2% | 19.4% | 47.6% | 22.7% | 0.2% |
| | Between the Fifth and Sixth Ring Roads | 2.35 | 1.7% | 7.3% | 37.5% | 53.4% | 0.1% |
| | Outside the Sixth Ring Roads | 1.84 | 1.1% | 7.1% | 34.9% | 56.7% | 0.2% |

Notes: [a] Including the emissions on the Fifth Ring Road; [b] Including the emissions on the Sixth Ring Road but excluding the emissions on the Fifth Ring Road

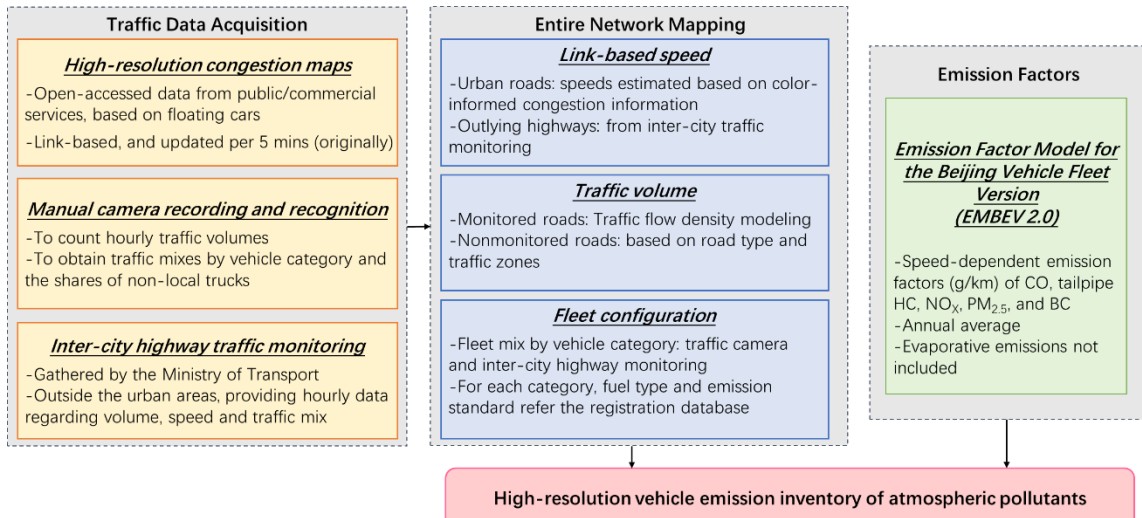

**Figure 1: A system diagram of the modeling methodology for EMBEV-link.**

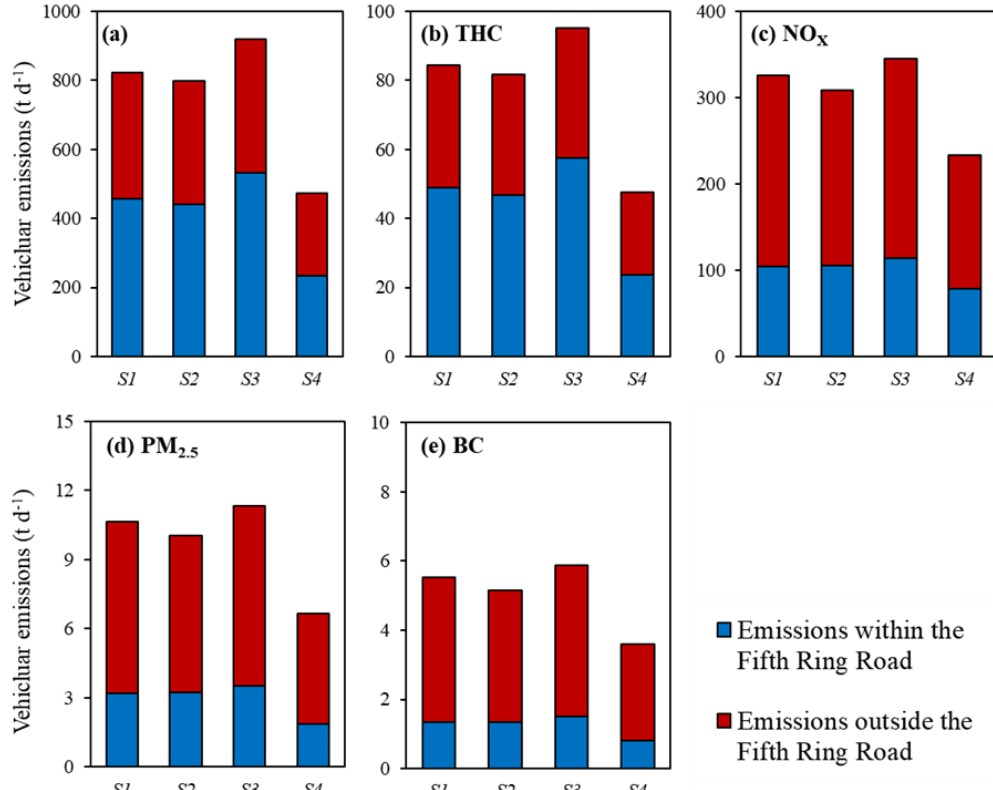

**Figure 2: Estimated total emissions under various traffic scenarios, S1 to S4: (a) CO, (b) THC, (c) NO$_X$, (d) PM$_{2.5}$ and (e) BC.**

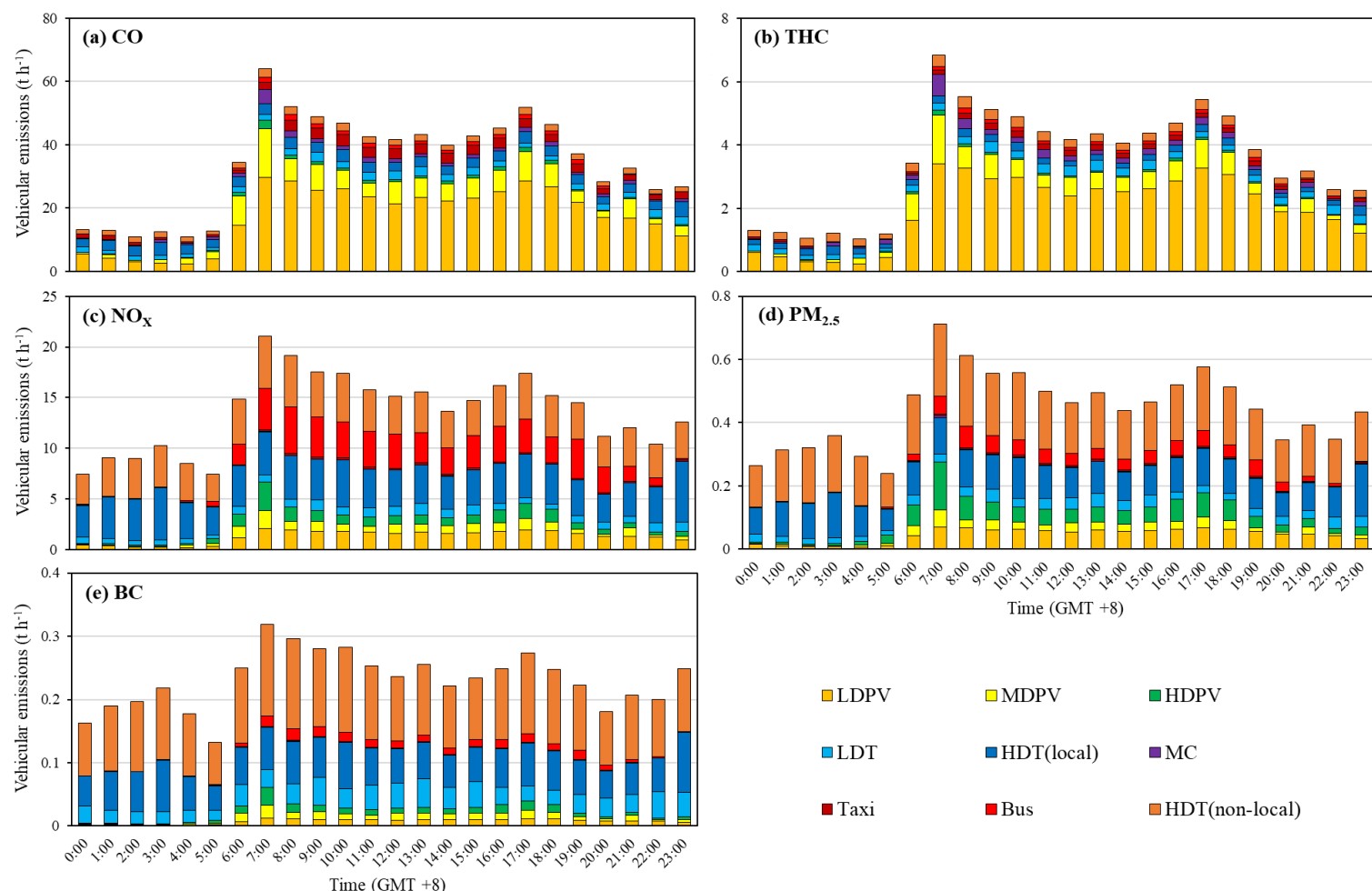

**Figure 3: Estimated hourly emissions by vehicle category under S1: (a) CO, (b) THC, (c) NOₓ, (d) PM₂.₅ and (e) BC**

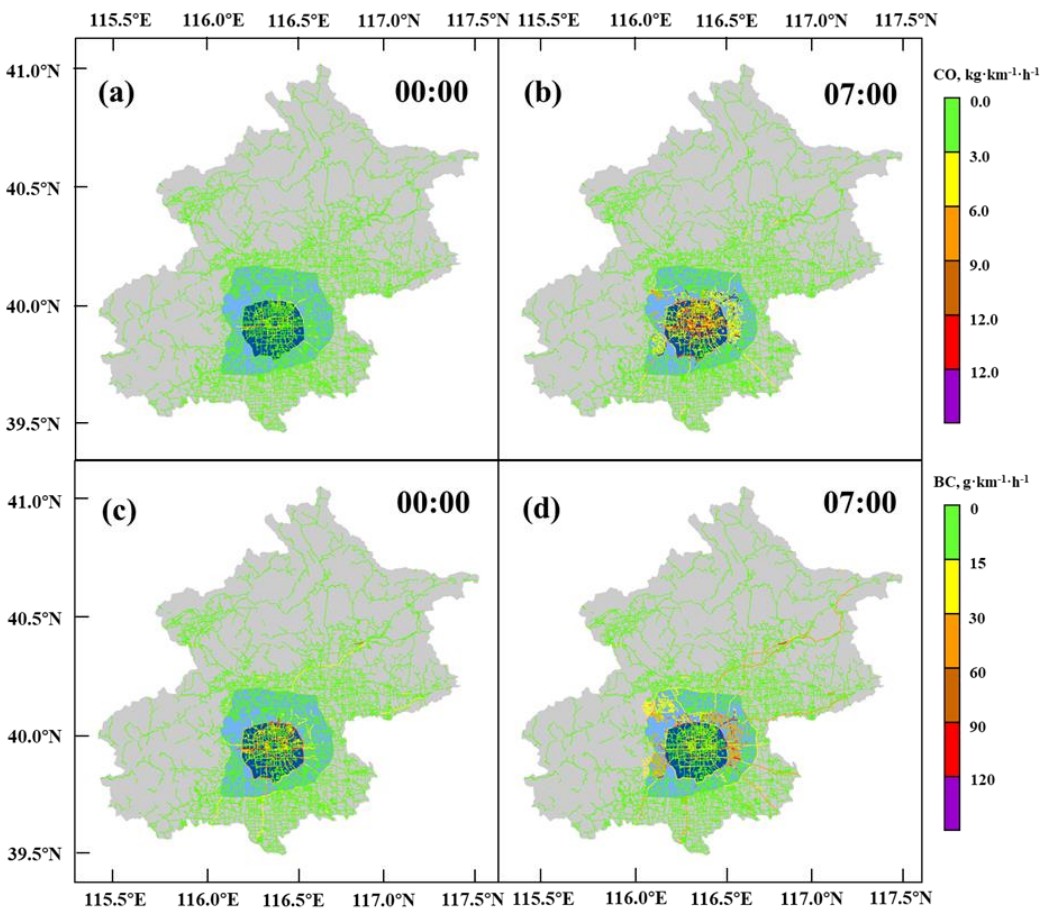

**Figure 4: Link-based emission intensity of CO (panels a and b) and BC (panels c and d) during a midnight hour (0:00 GMT+8) and**

**a morning rush hour (7:00 GMT+8).**

Note: Dark blue indicates the area within the Fifth Ring Road, while light blue indicates the area between the Fifth and Sixth Ring Roads.

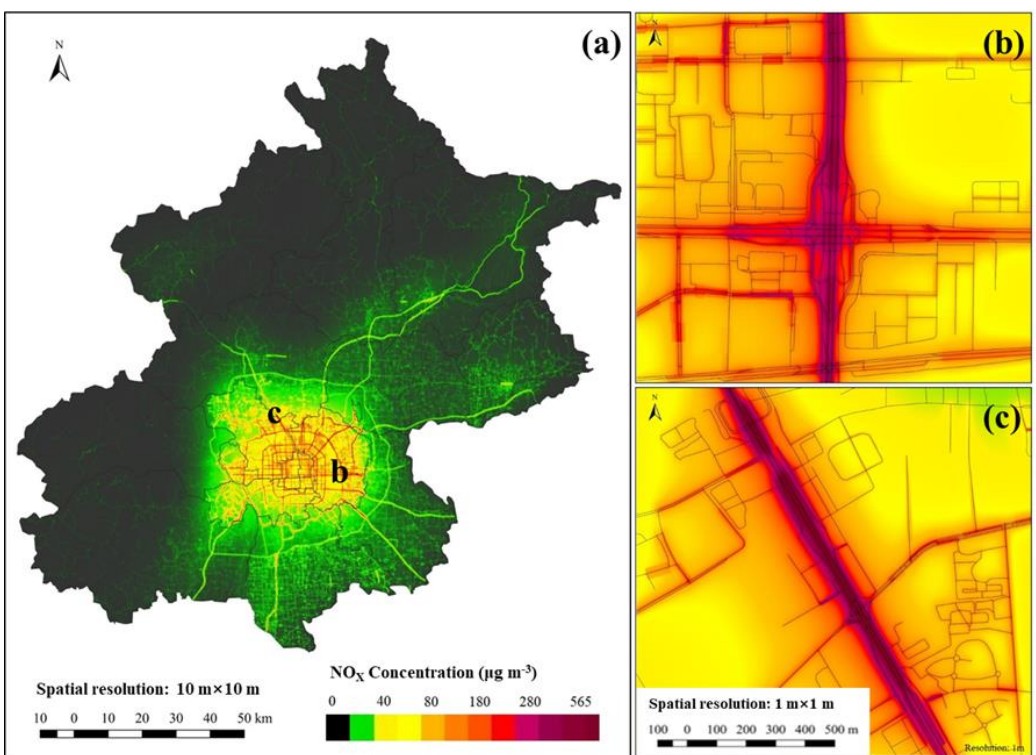

**Figure 5. High-resolution simulation of annual-average vehicular NO$_X$ concentrations for (a) the entire municipality, (b) Guomao**

**and (c) Xisanqi.**

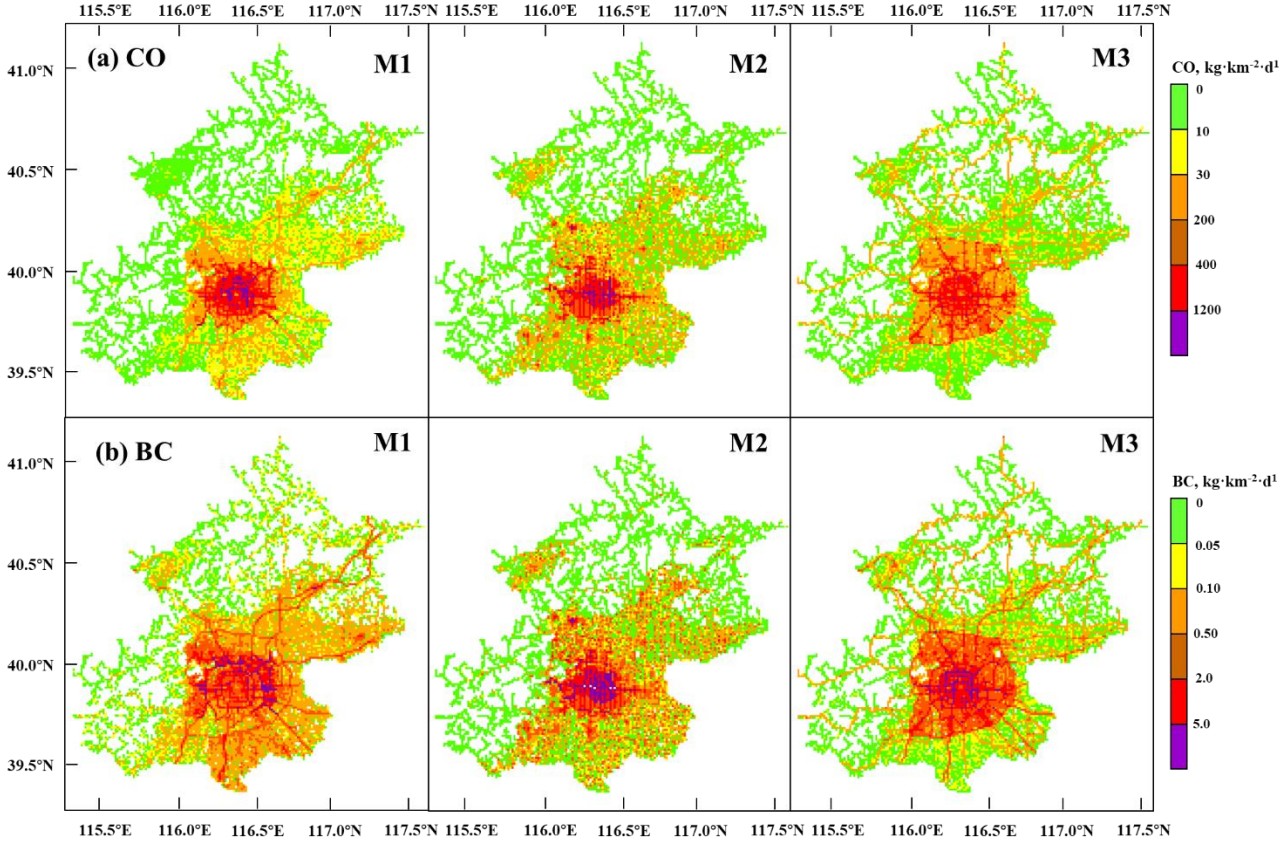

**Figure 6. Comparison of link-level emission intensity of (a) CO and (b) BC developed by various methods.**