# Peer review of "High-resolution mapping of vehicle emissions of atmospheric pollutants based on large-scale, real-world traffic datasets"

_Atmospheric Chemistry and Physics, 2019_

## Referee Comment (RC1) · Anonymous Referee #2 · 18 Mar 2019

This paper reports the results of a high resolution study of traffic emissions in the Beijing area. High resolution emission modeling such as that described in this paper combined with high spatial and temporal resolution ambient air quality measurements will become increasingly important in future urban atmospheric chemistry research. I have the following suggestions for improvement of the paper:

(1) The paper was difficult to read in places. It would benefit from careful proof reading by a native English speaker.

(2) It would be useful if the results were put into context in terms of impacts on human or ecosystem health. For example, in the abstract what is the significance of "high"

[Figure]

carbon monoxide and hydrocarbon emissions from traffic during rush hour. Do the CO and HC concentrations exceed air quality guidelines, and if so by how much, and how important is this?

(3) It appears that the NOx concentrations were estimated using a dispersion model without chemistry. If that is the case, the modeling estimation neglects losses of NOx via conversion to nitrate aerosol and HNO3. If so, the computed NOx concentrations are upper limits. Addition of clarification and an estimate of the overestimation if appropriate are needed.

(4) The results are based on traffic in 2013. My understanding is that there has been a large increase in the fleet size, and substantial decreases in the emissions from new vehicles, since 2013. These factors presumably offset to some degree. Discussion of the emission changes since 2013 resulting from changes to the vehicle fleet is needed.

(5) There are several examples of the use of non-scientific language in the paper. Terms such as "vivid", "soaring", "dramatically", "massive", "flooded", and "tales" should be replaced by more quantitative scientific text. The claim that "Beijing is a microcosm of other megacities" does not make sense.

---

## Referee Comment (RC2) · Anonymous Referee #3 · 9 Apr 2019

I've seen a variation of this paper before and must say that this version has much improved. The paper presents comprehensive analysis of on-road vehicle emissions in the metropolitan Beijing road network using a link based emission model, which is validated by an air dispersion modeling approach (note that there is no direct way of validating emission results). The four scenarios represent different policies that may affect the vehicle emission level in Beijing and thus have important policy implications. Overall I recommend the paper be considered for publication with this journal.

---

## Author Comment (AC1) · 19 Apr 2019

Reply to comments on "High-resolution mapping of vehicle emissions of atmospheric pollutants based on large-scale, real-world traffic datasets" by Daoyuan Yang et al.

*"Black" means the comments from reviewer and "Blue" text are our responses.*

We are deeply grateful to the reviewers of this paper for the very helpful comments. These comments are fully understood by the authors and individually responded to. Our responses to the comments are listed below, and major changes in the revised manuscript are also quoted in the response document.

**Reply to comments from Anonymous Referee #2:**

(1) The paper was difficult to read in places. It would benefit from careful proof reading by a native English speaker.

We have revised the specific comments by avoiding non-scientific wordings (i.e., according to Comment 5). We used the ACS Authoring Service to ensure the language quality (Certificate ID: 905A-3009-F9DC-30B3-E4EE). Before the final publication, further professional editing will be also conducted by the ACP editorial staff.

(2) It would be useful if the results were put into context in terms of impacts on human or ecosystem health. For example, in the abstract what is the significance of 'high' carbon monoxide and hydrocarbon emissions from traffic during rush hour. Do the CO and HC concentrations exceed air quality guidelines, and if so by how much, and how important is this?

The reviewer raised an important point regarding the potential risk of certain pollutants during traffic rush hours. In China, the National Ambient Air Quality Standards (NAAQS) set daily and hourly average CO concentration limits at 4 mg/m$^3$ and 10 mg/m$^3$. One-year observation records in 2013 for the most polluted site in Beijing (i.e., a traffic site near the South Third Ring Road) indicated that its maximum hourly concentration was 1.9 mg/m$^3$. Therefore, we conclude that the direct health impact from traffic CO emissions in Beijing would be not significant.

However, there is no such monitoring network in Beijing for certain HC species (e.g., particularly for mobile source air toxics, MSATs). For the reviewer's information, the average concentration of benzene in the urban area of Beijing was 1.7 μg/m$^3$ from a ten-day field study in the summer of 2012 (Li et al., 2014). This level was lower than the current annual limits in Japan (3 μg/m$^3$) and Europe (5 μg/m$^3$).

Li, L. et al., 2014, Pollution characteristics and health risk assessment of benzene homologues in ambient air in the northeastern urban area of Beijing, China. *J. Environ. Sci. 26* (1), 214-223.

(3) It appears that the $NO_X$ concentrations were estimated using a dispersion model without chemistry. If that is the case, the modeling estimation neglects losses of $NO_X$ via conversion to nitrate aerosol and $HNO_3$. If so, the computed $NO_X$ concentrations are upper limits. Addition of clarification and an estimate of the overestimation if appropriate are needed.

The reviewer understood correctly that we didn't include detailed atmospheric chemistry in the dispersion process of vehicular $NO_X$ emissions. In the comparison with ground-level observations, we only included the daytime $NO_X$ chemistry ($NO$-$NO_2$-$O3$) to estimate the in-situ $NO$ concentrations for the AQ sites.

Due to the lack of measurement data for city-scale $NO_Z$ concentrations, we relied on the atmospheric simulations to further quantify the potential impact from ignoring the production of daytime $NO_Z$. We referred to air quality simulation results conducted by using the WRF/CMAQ system on the 4-km scale (i.e., using the emission inventory data from Zheng et al., 2019). The results indicate that the daytime $NO_Z$ concentration would be 4.4 ppbv, which approximately 10% of concurrent $NO_X$ concentrations (see Page 10, Line 263 to 266). However, we are currently limited to further estimate the exclusive $NO_Z$ contribution from vehicular emissions in Beijing, which requires more advanced source apportionment tool.

Zheng, H., et al., 2019. Development of a unit-based industrial emission inventory in the Beijing–Tianjin–Hebei region and resulting improvement in air quality modeling. *Atmos. Chem. Phys.*, 19(6), 3447-3462.

 "*We acknowledge that the daytime concentration of other reactive oxides of nitrogen (i.e., $NO_Z$, including $HNO_3$ and $HONO$) could be approximately 10% of concurrent $NO_X$ concentrations by analyzing the air quality simulation outputs of Zheng et al. (2019). Further studies would be needed to couple dispersion and advanced atmospheric chemistry to better resolve urban pollution.*"

(4) The results are based on traffic in 2013. My understanding is that there has been a large increase in the fleet size, and substantial decreases in the emissions from new vehicles, since 2013. These factors presumably offset to some degree. Discussion of the emission changes since 2013 resulting from changes to the vehicle fleet is needed.

Beijing has adopted a series of license control and driving restriction polities to control excessively rapid growth in vehicle population. The total vehicle population during 2013-2017 only increased by 8%, which was much lower than the previous growth before 2010. Meanwhile, the Beijing Traffic Research Institute publishes annual traffic reports for the urban areas, which released annual-average urban traffic congestion by using an overall traffic index (from 0, least congested, to 10, most congested; different from the congestion index reported in the manuscript). The annual index values for 2013 to 2017 all ranged within 5.5 to 5.7, representing quite stable traffic conditions over the past years.

On the other hand, the fleet-average emission factors did receive significantly

reductions due to the newly implemented emission standards (China 5/V) and subsidized scrappage of older vehicles (Zhang et al., 2014). For example, compared with 2013, fleet-average emissions factors of CO, THC and $NO_X$ for light-duty gasoline vehicles were reduced by ~40%. With updated emission factors and actual traffic data for inter-city expressways in 2017, and scaled vehicle speeds according to the average traffic index, we estimate that daily emissions in 2017 weekdays are 523 tons for CO, 62.5 tons for THC, 256 tons for $NO_X$, 8.33 tons for $PM_{2.5}$ and 4.18 tons for BC, respectively. The emission reductions (~30% for CO and THC, and 20%~25% for $NO_X$, $PM_{2.5}$ and BC) relative to the 2013 levels would be majorly attributed to the improvements in average emission factors (see Page 8, Line 199 to 204).

Beijing Transport Institution (BTI), Beijing Transportation Annual Report (in Chinese), 2018. Available at http://www.bjtrc.org.cn/JGJS.aspx?id=5.2&Menu=GZCG.

Zhang, S., et al. 2014. Historic and future trends of vehicle emissions in Beijing, 1998–2020: A policy assessment for the most stringent vehicle emission control program in China. *Atmos. Environ.*, 89, 216-229.

*"The recent traffic monitoring data indicate the overall congestion in the urban area has not changed significantly, which is owing to the stringent restrictions on the registration of new vehicles in Beijing (BTI, 2018). On the other hand, average emission factors have decrease significantly due to the implementation of newer emission standards and the subsidized scrappage of older vehicles. As a result, we estimated that the total daily emissions would be 523 tons for CO, 62.5 tons for THC, 256 tons for $NO_X$, 8.33 tons for $PM_{2.5}$ and 4.18 tons for BC, respectively, in 2017. The significant reductions are primarily attributed to the improvements in average vehicle emission factors."*

(5) There are several examples of the use of non-scientific language in the paper. Terms such as "vivid", "soaring", "dramatically", "massive", "flooded", and "tales" should be replaced by more quantitative scientific text. The claim that "Beijing is a microcosm of other megacities" does not make sense.

Revisions are made in the revised manuscript.

**Reply to comments from Anonymous Referee #3:**

I've seen a variation of this paper before and must say that this version has much improved. The paper presents comprehensive analysis of on-road vehicle emissions in the metropolitan Beijing road network using a link based emission model, which is validated by an air dispersion modeling approach (note that there is no direct way of validating emission results). The four scenarios represent different policies that may affect the vehicle emission level in Beijing and thus have important policy implications. Overall I recommend the paper be considered for publication with this journal.

We appreciate the positive comments from the reviewer.

---

## Referee Report (RR1)

Yang et al. manuscript discussed an important issue of on-road mobile source emissions estimation based on the real-world traffic data. Overall, I think it is a good paper to be published after a few improvements. Based on my review, I would like to share a few comments and suggestions with authors.

1. While this paper covers well on on-road vehicle emissions from running exhaustion process, it lacks on addressing another critical vehicle emissions from evaporative processes which occur from on-road and off-network. Although authors mentioned in the manuscript that their study is limited to cover evaporative emissions due to the spatial coverage issue, it is very important to cover the evaporative emissions since they could contribute up to 30-40% of total emissions from vehicles depending on the regions. At least authors can provide more detail information on why they are estimating evaporative emissions other than spatial coverage issue.

2. Mobile source emissions, especially in a megacity like Beijing, are known to be a significant contributor of not only primary but secondary air pollution. Chemical characteristics of hydrocarbon emissions from mobile sources are very important to correctly understand secondary formation. I suggest authors to describe relationship between THC to NMVOCs at least, and more preferably include some level of discussion on major chemical species emissions by vehicle type.

3. It would be nice to see the flow diagram figure of EMBEV-Link (Core-programs/modules with inputs and outputs) to understand the model structure better.

4. It also shows why high-resolution activity data like hourly average speed, traffic density and VKT by each link can enhance the quality of hourly emissions. However, the hourly average speed calculated by congestion by link could also provide a biased speed-sensitive emissions since there should be more than a single speed value to present the traffic pattern during the congestion. The average speed would be most frequent speed during the traffic hour but there are higher and lower speed to be considered. Authors can easily find several journal papers that describe the importance of using average speed distribution factors by speed bins instead of a single average speed value. Although there is a merit to their approach to compute hourly average speed value, I think it should also mention about the future study on comparing the results against to the average speed distribution factors.

5. Although authors pointed out the importance of NOx emissions from HDT outside of Five Rings, it does not mention the sensitivity of diesel engine to local meteorological condition. It is a known fact that NOx from diesel vehicle shows a significant dependency to local ambient temperature and humidity. While covering this section will be out of scope of paper at this point, I think it is important for authors at least to mention its potential local meteorological condition impacts to NOx emissions

from HDT as well and what might be the impacts from it.

6. It is also not within the scope of this paper but it should consider to mention about how the EMBEV-Link can be updated to implemented to support photochemical modeling system other than a dispersion modeling system.

---

## Author Response (AR2)

Authors' Reply to comments on "High-resolution mapping of vehicle emissions of atmospheric pollutants based on large-scale, real-world traffic datasets"

*"Black" means the comments from reviewer and "Blue" text are our responses.*

The reviewer provided very candid and insightful comments on the manuscript. We fully understand that these comments represent the state-of-the-art directions in our research community. We have attempted to gather available measurements and utilize data analysis to address these comments. As noted by this reviewer, we expect the addition in this round of revision can better inform potential readers. However, some concerns (e.g., in particular, the temperature impact on diesel $NO_X$ emissions) are limited to be sophisticatedly addressed at this stage due to the lack of local measurement profiles. We will certainly continue to improve the EMBEV-Link by considering the comments as important future directions.

(1) While this paper covers well on on-road vehicle emissions from running exhaustion process, it lacks on addressing another critical vehicle emissions from evaporative processes which occur from on-road and off-network. Although authors mentioned in the manuscript that their study is limited to cover evaporative emissions due to the spatial coverage issue, it is very important to cover the evaporative emissions since they could contribute up to 30-40% of total emissions from vehicles depending on the regions. At least authors can provide more detail information on why they are estimating evaporative emissions other than spatial coverage issue.

The original EMBEV model did include the evaporative emissions by referring to global measurement/modeling results (Zhang et al., 2014). Later on, more local measurements of evaporative emissions became available by using SHED tests (Liu et al., 2015). By considering these SHED results (diurnal and hot soak emissionss) and the running loss rates used in the MOVES model (EPA, 2012; note: we used the average of Tier-1 (ORVR-equipped) and pre-Tier 1 vehicles), we estimated that the average daily evaporative emissions in 2013 are 32.3 tons of THC. The estimated daily evaporative emissions are approximately responsible for 28% of total THC emissions (i.e., 31% of total THC emissions from gasoline vehicles). Yes, the proportion well fits the reviewer's estimate. However, as we noted in the manuscript, we are limited to quantify the detailed spatial and temporal distributions for such off-network emissions. The uncertainty in actual running loss rates and the seasonal variability in evaporative emissions should be paid attention to. We revised our manuscript by adding the estimated total daily emissions of evaporative THC (see Page 4 Line 104 to Line 107).

References:

U.S. EPA, 2012. Development of Evaporative Emissions Calculations for the Motor Vehicle Emissions Simulator MOVES2010, EPA-420-R-12-027, United States Environmental Protection Agency, Washington, DC, available at: https://nepis.epa.gov/Exe/ZyPDF.cgi/P100F3ZY.PDF?Dockey=P100F3ZY.PDF.

Liu H. et al., 2015. VOC from Vehicular Evaporation Emissions: Status and Control Strategy. *Environ. Sci. & Tech.*, 49(24), 14424.

Zhang S. et al, 2014. Historic and future trends of vehicle emissions in Beijing, 1998–2020: A policy assessment for the most stringent vehicle emission control program in China, *Atmos. Environ.*, 89, 216-229.

(2) Mobile source emissions, especially in a megacity like Beijing, are known to be a significant contributor of not only primary but secondary air pollution. Chemical characteristics of hydrocarbon emissions from mobile sources are very important to correctly understand secondary formation. I suggest authors to describe relationship between THC to NMVOCs at least, and more preferably include some level of discussion on major chemical species emissions by vehicle type.

The species-resolved profiles are important to improve air quality simulations, for example, to be incorporated with atmospheric chemical mechanisms of ozone and secondary organic aerosol formations. Because the fuel properties (e.g., contents of aromatics and olefins) in China are different from those in the U.S. and Europe, we opted to use local measurement data to answer this question. We consider NMHC and NMVOC according to the U.S. EPA's definitions (test methods noted in brackets):

NMHC = THC (FID) - $CH_4$ (FID)

NMVOC = NMHC + aldehyde/ketone (HPLC) - acetone (HPLC) – ethane (GC-MS)

In the past two years, we employed dynamometer and analytical instruments (GC-MS and HPLC) to quantify the chemical compositions of major tailpipe VOC species. We have measured nearly ten light-duty gasoline vehicles in China, which comply with China 2 to China 5 emission standards (Wu et al., 2019). The preliminary results indicated that NMHC could account for 85% to 90% of THC emissions. No significant difference could be observed among various emission standard categories. Further considering the presence of aldehyde/ketone, NMVOC could contribute to 88% to 95% of total THC emissions. However, we have not analyzed VOC species from diesel emissions (measurements are ongoing). Since gasoline vehicles are responsible for more than 90% of total THC emissions in Beijing, we can conclude that about 90% of THC emissions should be NMVOC emissions.

For certain VOC species of particular concerns, we did observe that the proportions in THC emissions could be impacted by their emission standards (the same project of Wu et al., 2019). Taking BTEX (benzene, toluene, ethylbenzene, and xylene) for example, based on our tested vehicles, we've observed that the proportions in THC emissions increased from 4% for China 2 and China 3 vehicles to 20% for China 5 vehicles. We will report the chemical characteristics of VOC emissions (e.g., species-resolved profiles, ozone formation potentials, SOA yield potentials) in our future papers, since they are beyond the scope of this study. In the manuscript, we have added the explanations of NMVOC emissions for potential users of air quality models (see Page 4 Line 109 to Line 113).

We've added one flow diagram (Fig. 1) for the EMBEV-Link modeling system in the revised manuscript according to the reviewer's comment (see Page 3 Line 85 to Page 4 Line 87).

[Figure]

Figure 1: A system diagram of the modeling methodology for EMBEV-link.

(4) It also shows why high-resolution activity data like hourly average speed, traffic density and VKT by each link can enhance the quality of hourly emissions. However, the hourly average speed calculated by congestion by link could also provide a biased speed-sensitive emissions since there should be more than a single speed value to present the traffic pattern during the congestion. The average speed would be most frequent speed during the traffic hour but there are higher and lower speed to be considered. Authors can easily find several journal papers that describe the importance of using average speed distribution factors by speed bins instead of a single average speed value. Although there is a merit to their approach to compute hourly average speed value, I think it should also mention about the future study on comparing the results against to the average speed distribution factors.

We agree with the reviewer that a few researchers note the useful features by using a distribution of average speeds rather than one single speed input (e.g., Smit et al.,

2008). We would like to address the reviewer's comment in two aspects.

First, as the Supplement illustrates, although the congestion information was updated per 5 mins, the instantaneous congestion index represents quite a wide interval of traffic speeds (e.g., see Table S3). Thus, using instantaneous or quasi-instantaneous (e.g., per 10 mins or 15 mins) traffic congestion data to estimate real-time speeds could lead to substantial uncertainties. At this stage, we don't have other data sources to validate speed estimation in such short temporal duration. Thus, we are not able to develop reliable speed distributions within the hourly basis. Therefore, we opt to use hourly speed as a reasonable temporal resolution.

Second, our EMBEV model is a dynamic system to automate real-time emission calculation in Beijing. For example, we've obtained available congestion maps from more than 300 workdays in 2013-2014. Taking West Third Ring Rd. (urban expressways) and Zizhuqiao Rd. (sub-arterial road close to West Third Ring Rd.) for example, we plotted their hourly speed distributions during the morning rush hour (8:00 GMT+8), and further estimated the distributions of hourly CO emission factors (g km$^{-1}$) for LDPVs and hourly CO emission intensities for the total vehicles (g km$^{-1}$ h$^{-1}$) (see Fig. S8). The 95% variation intervals of CO emission intensities are estimated as to 29.8~36.4 g km$^{-1}$ h$^{-1}$ for West Third Ring Rd. (Fig. S8c) and 3.24~4.22 g km$^{-1}$ h$^{-1}$ for Zizhuqiao Rd. (Fig. S8f). Yet, we do confirm that the average hourly speeds used in our EMBEV model in the workday scenario (*S1*) will not lead to significant bias for emission estimations. We have added one figure (see Supplementary Information Figure S8) and one paragraph (see Page 9 Line 248 to Line 254) to discuss the issue.

[Figure]

Figure S8. The distributions of workday hourly speeds (panels a and d for West Third Ring Rd. and Zizhuqiao Rd., respectively), CO emission factors for LDPVs (panels b and e), and CO emission intensity for total vehicles (panels c and f) during a typical morning rush duration (8:00 GMT+8).

Refence:

Smit R., et al., 2008. Do air pollution emissions and fuel consumption models for roadways include the effects of congestion in the roadway traffic flow? *Environ. Modell. & Softw.*, 23 (10-11), 1262-1270.

Wu X, et al., 2019. Assessment of ethanol blended fuels for gasoline vehicles in China: fuel economy, regulated gaseous pollutants and particulate matters. *Environ. Pollut.*, under review.

(5) Although authors pointed out the importance of NOx emissions from HDT outside of Five Rings, it does not mention the sensitivity of diesel engine to local meteorological condition. It is a known fact that NOx from diesel vehicle shows a significant dependency to local ambient temperature and humidity. While covering this section will be out of scope of paper at this point, I think it is important for authors at least to mention its potential local meteorological condition impacts to NOx emissions from HDT as well and what might be the impacts from it.

Characterizing the seasonal variability of $NO_X$ emissions is essential to understand a few important atmospheric issues (e.g., near-road $NO_2$ concentrations, wintertime nitrate formation). In the revised manuscript, we introduce that the recent remote sensing results in UK (Grange et al., 2019) and in other European countries (e.g., the CONOX project) (Borken-Kleefled and Dallmann, 2018) have identified strong temperature dependence for NOx emissions from diesel cars. Their $NO_X$ emissions could be significantly increased under low temperature conditions than the normal conditions (~20 ℃), which has not been reflected by current emission models (e.g., MOVES). However, diesel engines are mostly applied by heavy-duty vehicles in China. We expect that the trend in temperature dependence for $NO_X$ emissions would also exist among HDDVs, but we are limited to develop detailed corrections due to the lack of usable measurement data. We have added one paragraph (see Page 10 Line 281 to Page 11 Line 288) to discuss the issue.

Grange, S. K. et al., 2019. Strong Temperature Dependence for Light-Duty Diesel Vehicle NOx Emissions. *Environ. Sci. & Tech.*.

Borken-Kleefeld, J.; Dallmann, T., 2018. Remote sensing of motor vehicle exhaust emissions. White Paper. International Council on Clean Transportation, Washington, DC.
https://www.theicct.org/sites/default/files/publications/Remote-sensing-emissions_ICCT-White-Paper_01022018_vF_rev. pdf 2018.

(6) It is also not within the scope of this paper but it should consider to mention about how the EMBEV-Link can be updated to implemented to support photochemical modeling system other than a dispersion modeling system.

Very nice point. Actually, several projects regarding regional and city-level air quality simulations by using chemical transport models (WRF/CMAQ) are undergoing with our EMBEV-Link as emission input. We have added one paragraph (see Page 13 Line 351 to Line 357) to discuss the issue.

[revised manuscript text omitted]